# Central But Not General Obesity Is Positively Associated with the Risk of Hyperhomocysteinemia in Middle-Aged Women

**DOI:** 10.3390/nu11071614

**Published:** 2019-07-16

**Authors:** Yingying Wang, Yonggen Jiang, Na Wang, Meiying Zhu, Xing Liu, Ruiping Wang, Feng Jiang, Yue Chen, Qi Zhao, Genming Zhao

**Affiliations:** 1Department of Epidemiology, School of Public Health, Fudan University, NO 130, Dongan Road, Xuhui District, Shanghai 200032, China; 2Key Laboratory of Public Health Safety of Ministry of Education, Shanghai 200032, China; 3Songjiang District Center for Disease Control and Prevention, Shanghai 201620, China; 4School of Epidemiology and Public Health, Faculty of Medicine, University of Ottawa, Ottawa, ON K1G 5Z3, Canada

**Keywords:** homocysteine, central obesity, menopause, cardiovascular disease

## Abstract

Objective: Obesity and homocysteine (Hcy) are two important risk factors for cardiovascular disease (CVD). However, evidence on the association between obesity and Hcy concentration was conflicting. The aim of our study is to explore the associations of general and central obesity with hyperhomocysteinemia (HHcy) in middle-aged women. Methods: The current analysis was based on data from 11,007 women aged 40–60 years. Height, weight, and waist circumference (WC) were measured and serum homocysteine was determined. Multiple logistic regression models were used to assess the associations of the risk of hyperhomocysteinemia (HHcy, Hcy > 15 μmol/L) with BMI and WC. Results: 13.71% women had HHcy. The prevalences of BMI-based general obesity and WC-based central obesity were 11.17% and 22.88%, respectively. Compared with non-obese women, the mean serum Hcy concentration was significantly higher in WC-based central obese women (*p* = 0.002), but not in BMI-based general obese women (*p* > 0.05). In the multiple logistic regression models, central obesity was positively related to the risk of HHcy (OR = 1.30, 95% CI = 1.10 to 1.52), while general obesity was inversely related to the risk of HHcy (OR = 0.82, 95% CI = 0.72 to 0.93 and OR = 0.71, 95% CI = 0.57 to 0.89). Conclusions: Central obesity was positively related to the risk of HHcy, while general obesity was negatively related. Menopause showed no effect modification on these associations.

## 1. Introduction

Homocysteine (Hcy) is a sulfurous α-amino acid which acts as an intermediate molecule in methionine metabolism [1]. Hyperhomocysteinemia (HHcy), featured by a level of Hcy above 15μmol/L in the blood, has been identified as an independent risk factor for cardiovascular disease (CVD). Many studies have demonstrated that HHcy is linked to endothelial damage [2], atherosclerosis [3], and venous thrombosis [4]. Obesity, defined as abnormal and excess fat accumulation due to an imbalance between calorie intake and expenditure, is also a leading cause of CVD and some other chronic conditions [5]. Two patterns of obesity, which are general (peripheral) obesity and central (abdominal) obesity, are often evaluated by Body Mass Index (BMI) and Waist Circumference (WC) respectively.

Evidence on the association between BMI and Hcy concentration in the general population has been conflicting [6,7,8], and there are limited data for women [9,10]. Middle-aged women are more likely to accumulate fat due to a decline in basal metabolic rate and lack of physical activities, [11] and estrogen deprivation secondary to menopause causes several adverse cardio-metabolic consequences that are correlated with obesity [12]. We speculated that menopause may contribute to the influences of obesity on Hcy. In this cross-sectional study conducted in East China, we explored the association between obesity and the risk of HHcy in middle-aged women and assessed the potential effect modification of menopause on this association.

## 2. Materials and Methods

### 2.1. Study Design and Subjects

A community-based cross-sectional study was conducted in Songjiang District of Shanghai, China, from June 2016 to December 2017. A multistage cluster sampling was employed to recruit subjects. Four communities (Zhongshan, Xinqiao, Sheshan and Maogang) and all residents aged 20–74 years who had been living in Songjiang District for at least 5 years were enrolled. All eligible subjects were approached by trained interviewers using a structured questionnaire to collect information on sociodemographic, physical activity and chronic disease history. Menstrual and reproductive history was collected for all women. Anthropometric measurements and biological sample collections were performed at the same time. Informed written consents were obtained from all the participants, and the study was approved by the ethical review board of the School of Public Health of Fudan University. The current analysis included a total of 11,007 women aged 40–60 years with complete data of the Hcy level and anthropometric values (Figure 1).

### 2.2. Anthropometric and Other Measurements

Height (cm), weight (kg) and waist circumferences (cm) of the waist and hip were measured by trained health professionals according to a standard protocol. Height and weight were respectively measured to the nearest 0.1 cm and 0.1 kg, with the subjects standing without shoes and wearing light clothing only. Waist circumference (WC) was measured to 0.1 cm at the midpoint between the lower rib and the upper iliac crest. Body mass index (BMI, kg/m^2^) was calculated as the weight in kilograms divided by the square of height in meters. All subjects were grouped into three BMI categories (normal weight: ≤23.9 kg/m^2^; overweight: 24.0–27.9 kg/m^2^; and obesity: ≥28.0 kg/m^2^) and into two WC categories (normal weight: <85.0 cm; and central obesity: ≥85.0 cm) according to a Chinese Guideline [13]. Menopause was defined as the permanent cessation of menstruation for ≥12 consecutive months due to ovarian reserve depletion, without any history of surgical or iatrogenic menopause. Exercising was defined as doing physical exercises like walking or sports activities like ball games for at least 10 min every week over the past year. Tea drinking was defined as drinking tea at least three times per week for more than 6 consecutive months.

### 2.3. Biological Sample Collection and Determination

Blood samples (approximately 10 mL) were collected through antecubital vein puncture after a 12-h overnight fast from each subject. Serum and plasma samples were separated, and then kept at −80 °C freezer until transported to the DiAn medical laboratory center for analysis. Serum Hcy was determined by using an enzymatic cycling method, [14] and serum creatinine was determined using enzymatic analysis [15]. A subject with a Hcy level of greater than 15 mmol/L was considered to have Hhcy [1].

### 2.4. Statistical Analysis

We used Student’s t test and one-way ANOVA for comparisons of continuous data and χ^2^ tests for categorical data. Multiple logistic regressions were used to determine the associations of HHcy with BMI and WC, and adjusted odd ratios (ORs) and 95% confidence intervals (95% CIs) were calculated taking potential confounding and effect modification into account. Covariables were considered into the analysis included age, menopausal status, CVD-related comorbidities (including hypertension, coronary heart diseases, stroke, diabetes, hyperlipidemia, fatty liver, chronic nephrosis and cancer), education level, occupational status, exercising, and tea drinking. Smoking (0.28%) and regular alcohol drinking (0.73%) were not common among the women subjects and were not considered in the analysis. The level of statistical significance was defined as α = 0.05 of two-side probability. All analysis was performed by using SPSS software for Windows (version 24.0, IBM Crop., Armonk, NY, USA).

### 2.5. Ethics Statement

The study was approved by the Institute Review Board (IRB) of School of Public Health, Fudan University and the IRB of SCH. Written informed consent was obtained from all subjects in the study before enrollment. (Authorization number: IRB#2016-04-0586-S).

## 3. Results

Of 11,007 women aged 40–60 years, there were 1509 (13.71%) women with HHcy. In all, 1229 (11.17%) were generally obese based on BMI, and 2518(22.88%) were centrally obese based on WC (Table 1). Women with HHcy were more likely to be older and postmenopausal (*p* < 0.001), and had a higher proportion of central obesity (*p =* 0.002) as compared with those without HHcy. However, the proportion of general obesity was comparable between women with and without HHcy (*p* = 0.590). Women with HHcy were more likely to have cardiovascular-related diseases, be retired, and were less likely to do exercise or to drink tea (*p* < 0.001).

The mean serum Hcy concentration was significantly higher in women with WC-based central obesity (12.09 ± 4.11 μmol/L) than without (11.79 ± 4.38 μmol/L) (*p* = 0.002), but showed no significant difference between BMI groups. The Hcy concentration increased after menopause in all the study groups (Figure 2).

There was a positive association between central obesity measured by WC and the risk of HHcy (OR = 1.30, 95% CI = 1.10 to 1.52) (Table 2). This association was not significantly modified by either age or menopausal status (*p*-interaction > 0.05). On the contrary, BMI tended to be negatively related to the risk of HHcy (Table 3). Those with overweight had a lower risk of HHcy (OR = 0.88, 95% CI = 0.78 to 0.99) than those with normal weight after adjustment for potential confounders (Model 2). After additional adjustment for WC in Model 3, the pooled ORs were 0.82 (95% CI = 0.72 to 0.93) for overweight women and 0.71 (95% CI = 0.57 to 0.89) for obese women, as compared with those with normal weight.

## 4. Discussion

In this large cross-sectional study of 11,007 middle-aged women in rural east China, we observed that central obesity was positively related to the risk of HHcy, while general obesity was negatively related to HHcy. Women with central obesity had a higher concentration of Hcy and a higher risk of HHcy than women with normal weight, which is consistent with previous studies [16,17]. The level of Hcy, as a crucial intermediate of the methionine cycle, elevates when the rate of Hcy production exceeds the capacity of the reaction that convert Hcy back to methionine (remethylation) or to cysteine (transulphuration). Both remethylation and transulphuration depend on folate and vitamin B [18]. Obese women are likely to have constantly low plasma folate concentration affected by body composition and changes in plasma volume [19]. On the other hand, some exogenous factors including obesity and physical inactivity contribute to the deficiency of cofactors essential to Hcy metabolism [20]. Methionine has been confirmed as the most important methyl donor, and the disorder of methionine metabolism contributes to fatty liver, which is significantly and positively related to central obesity [21].

In addition, Bjorck et al. found that Hcy was significantly associated with both serum insulin and homeostasis model assessment (HOMA) of insulin resistance (IR) in a population-based sample of Swedish [22]. Levels of serum insulin play a role in homocysteine metabolism, possibly through influencing glomerular filtration or activity of key enzymes including 5,10-methylenetetrahydrofolate reductase or cystathione B-synthase [23]. Obesity, especially central obesity, is a risk factor for the development of IR. Thus, an indirect relationship may exist between the Hcy levels and central obesity [24].

We observed no significant association between BMI and Hcy concentrations, and a tendency BMI negatively associated with HHcy. Amany et al. also reported that Hcy was negatively related to BMI, but only after adjustment for cysteine [7]. It has been generally recognized that elevated BMI contributes to the pathogenesis of related diseases. The inverse relationship observed in our study may also be phenomenon of ‘obesity paradox’, [25] which exists due to methodological misunderstandings. BMI fails to distinguish between fat tissue and lean mass. Sat et al. revealed that an increase in total fat tissue percent is associated with an increase in Hcy concentrations [26]. Loss of lean mass, especially skeletal muscle, the largest insulin-responsive target tissue, also causes IR [27]. Thus, BMI as a proxy for general obesity is less sensitive than WC for central adipose in predicting chronic diseases [28]. Large prospective studies reported a greater risk of mortality in the group with the lowest BMI but also a higher WC level [29,30]. In addition, a cohort study in which participants were followed up for 27 years indicated that BMI variability was not associated with CVD risk among individuals with or without obesity [31].

The mid-life period is a pivotal window for weight gains, which reaches its natural maximum at the onset of menopause [11]. In China, the prevalence of overweight, obesity and central obesity was 34.6%, 13.0%, and 24.6% in women aged 40–49 years, and 38.1%, 15.2% and 38.4% in women aged 50–59, respectively [32]. This condition is primarily due to chronological aging [33] and ovarian aging [34]. The negative associations between BMI and HHcy were only presented in older women, postmenopausal women or ones with CVD-related comorbidities. The interval between menarche and menopause determines a woman’s natural reproductive span. The longer reproductive span a woman experiences, the higher estrogens she accumulates [35]. Decline in estrogens accelerate aging and occurrence of some chronic diseases. Women are subject to perimenopausal syndromes at early menopause transition, like insomnia, vasomotor and hot flashes. We speculate that these syndromes may trigger metabolic disorders and influence Hcy level.

Strengths of the study included the large sample size and its recruitment from community-based population. To our knowledge, this is the first study to investigate the effects of different obesity types on HHcy among women at perimenopausal period. Our study also has some limitations. This cross-sectional survey was conducted only in the Songjiang District of Shanghai. There were certain limitations in the extrapolation of the study results and as a result the incidence level of HHcy in middle-aged women in Songjiang District could be not estimated. Detailed information on folate and vitamin B status were not available for all subjects. More accurate measurements were not performed for body composition, like dual-energy X-ray absorptiometry (DEXA) or computed tomography (CT) scans. Therefore, follow-up and corresponding examinations should be strengthened in future studies to obtain more accurate epidemiological data.

## 5. Conclusions

Central obesity was positively related to the risk of HHcy in a cross-sectional study, while general obesity was negatively related. WC seems to be a more reasonable index to assess the effect of obesity on HHcy and potential CVD risk as compared with BMI. Obesity may influence the risk of HHcy independently from menopause.

## Figures and Tables

**Figure 1 nutrients-11-01614-f001:**
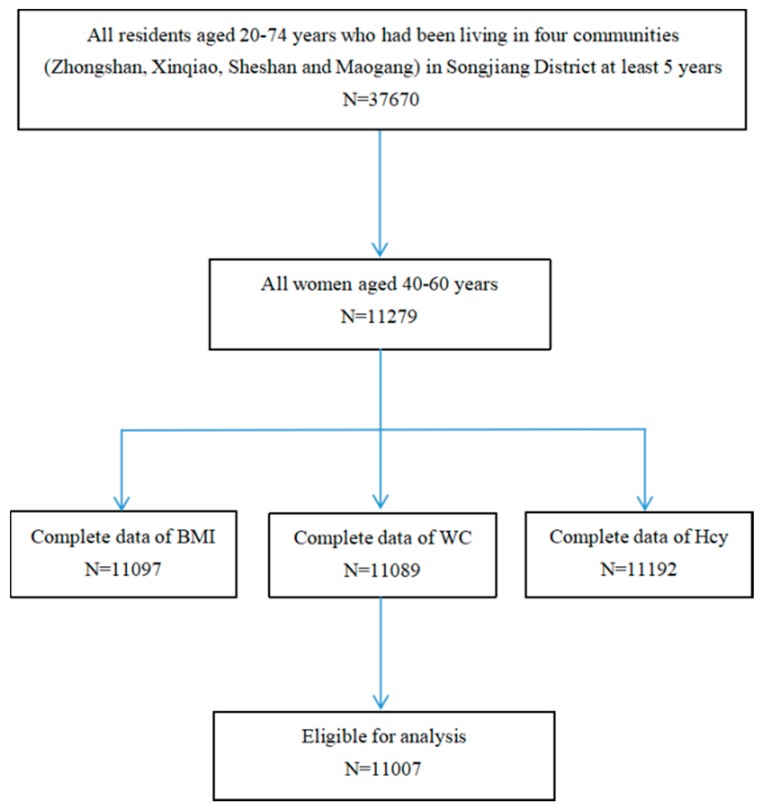
Flow chart for the study.

**Figure 2 nutrients-11-01614-f002:**
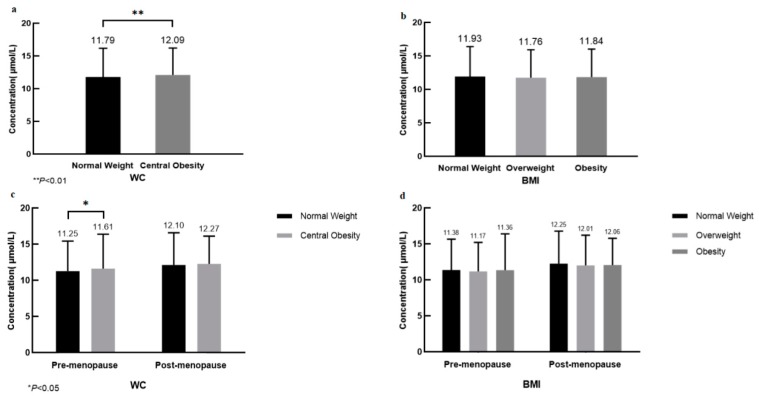
The concentrations of Hcy according to waist circumference (WC) or body mass index (BMI). (**a**) Subjects were divided into two WC categories (normal weight: <85.0 cm; and central obesity: ≥85.0 cm); (**b**) Subjects were divided into three BMI categories (normal weight: ≤23.9 kg/m^2^; overweight: 24.0–27.9 kg/m^2^; and obesity: ≥28.0 kg/m^2^); (**c**) and (**d**) Stratified by menopause for further analysis.

**Table 1 nutrients-11-01614-t001:** Distributions of study variables among participants with and without hyperhomocysteinemia (HHcy).

	All Subjects (*n* = 11,007)	Non-Hhcy (*n* = 9498)	Hhcy (*n* = 1509)	*p* Value
**Age (years)**				<0.001
40–45	1211 (11.00)	1097 (11.55)	114 (7.55)	
46–50	2225 (20.21)	1997 (21.03)	228 (15.11)	
51–55	4057 (36.86)	3504 (36.89)	553 (36.65)	
56–60	3514 (31.93)	2900 (30.53)	614 (40.69)	
**Menopausal status**				<0.001
Pre-menopause	3717 (33.77)	3324 (35.00)	393 (26.04)	
Post-menopause	7290 (66.23)	6174 (65.00)	1116 (73.96)	
**Body mass index (kg/m^2^)**				0.590
<24.0 (normal weight)	5807 (52.76)	4993 (52.57)	814 (53.94)	
24.0–27.9 (overweight)	3971 (36.08)	3443 (36.25)	528 (34.99)	
≥28.0 (obesity)	1229 (11.17)	1062 (11.18)	167 (11.07)	
**Waist circumference (cm)**				0.002
<85.00 (normal weight)	8489 (77.12)	7372 (77.62)	1117 (74.02)	
≥85.00 (central obesity)	2518 (22.88)	2126 (22.38)	392 (25.98)	
**Cardiovascular-related comorbidities**				<0.001
No	7398 (67.21)	6451 (67.92)	947 (62.76)	
Yes	3609 (32.79)	3047 (32.08)	562 (37.24)	
**Education**				0.028
Middle school or below	3889 (35.33)	3318 (34.93)	571 (37.84)	
High school or above	7118 (64.67)	6180 (65.07)	938 (62.16)	
**Retired**				<0.001
No	5162 (46.90)	4543 (47.83)	619 (41.02)	
Yes	5845 (53.10)	4955 (52.17)	890 (58.98)	
**Exercising**				0.013
No	7442 (67.61)	6380 (67.17)	1062 (70.38)	
Yes	3565 (32.39)	3118 (32.83)	447 (29.62)	
**Smoking**				0.359
No	10,967 (99.64)	9467 (99.67)	1500 (99.40)	
Yes	31 (0.28)	25 (0.26)	6 (0.40)	
**Alcohol drinking**				0.105
No	10,927 (99.27)	9424 (99.22)	1503 (99.60)	
Yes	80 (0.73)	74 (0.78)	6 (0.40)	
**Tea drinking**				0.004
No	9781 (88.86)	8407 (88.51)	1374 (91.05)	
Yes	1226 (11.14)	1091 (11.49)	135 (8.95)	

**Table 2 nutrients-11-01614-t002:** Odds ratios (ORs) and 95% confidence internals (CIs) for HHcy associated with waist circumference.

	Total	Normal Weight		Central Obesity	
		Cases/n	OR (ref)	Cases/n	OR (95% CI)
**All subjects ^a^**	11,007	1117/8491		392/2518	
Model 1			1.00		1.13 (1.00–1.29)
Model 2			1.00		1.10 (0.97–1.26)
Model 3			1.00		1.30 (1.11–1.52) **
**Age (years) ^b^**			1.00		
40–50	3436	279/2906	1.00	63/530	1.51 (1.04–2.20) *
51–60	7571	838/5583	1.00	329/1988	1.27 (1.06–1.51) **
**Menopausal status ^b^**					
Pre-menopause	3717	305/3050	1.00	88/667	1.44 (1.03–2.01) *
Post-menopause	7290	812/5439	1.00	304/1851	1.26 (1.05–1.51) *
**CVD-related comorbidities ^b^**					
No	7398	755/6074	1.00	191/1324	1.17 (0.94–1.44)
Yes	3610	361/2416	1.00	201/1194	1.52 (1.20–1.94) **

**^a^** Model 1 Adjusted for age only. Model 2 Adjusted for age, menopausal status, CVD-related comorbidities, education, retired, exercising, tea drinking, and serum creatinine. Model 3 Additionally adjusted for body mass index (BMI). **^b^** Variables included in the model were age, menopausal status, CVD-related comorbidities, education, retired, exercising, tea drinking, serum creatinine, and BMI. * 0.01 < *p* < 0.05; ** *p* < 0.01.

**Table 3 nutrients-11-01614-t003:** Odds ratios (ORs) and 95% confidence internals (CIs) for HHcy associated with body mass index.

	Total	Normal Weight		Overweight		Obesity	
		Cases/n	OR (ref)	Cases/n	OR (95%CI)	Cases/n	OR (95%CI)
**All subjects ^a^**	11,007	814/5807		528/3971		167/1229	
Model 1			1.00		0.89 (0.79–1.00)		0.91 (0.76–1.09)
Model 2			1.00		0.88 (0.78–0.99) *		0.87 (0.73–1.05)
Model 3			1.00		0.82 (0.72–0.93) **		0.71 (0.57–0.89) **
**Age (years) ^b^**							
40–50	3436	213/2090	1.00	99/1038	0.86 (0.65–1.13)	30/308	0.74 (0.45–1.23)
51–60	7571	601/3717	1.00	429/2933	0.81 (0.70–0.94) **	137/921	0.70 (0.54–0.90) **
**Menopausal status ^b^**							
Pre-menopause	3717	231/2162	1.00	117/1167	0.80 (0.61–1.03)	45/388	0.74 (0.47–1.15)
Post-menopause	7290	583/3645	1.00	411/2804	0.82 (0.71–0.96) *	122/841	0.70 (0.54–0.90) **
**CVD-related comorbidities ^b^**							
No	7398	559/4360	1.00	308/2434	0.90 (0.77–1.06)	80/604	0.89 (0.66–1.20)
Yes	3609	255/1447	1.00	220/1537	0.67 (0.54–0.84) **	87/625	0.52 (0.37–0.73) **

**^a^** Model 1 Adjusted for age only. Model 2 Adjusted for age, menopausal status, CVD-related comorbidities, education, retired, exercising, tea drinking, and serum creatinine. Model 3 Additionally adjusted for waist circumference. **^b^** Variables included in the model were age, menopausal status, CVD-related comorbidities, education, retired, exercising, tea drinking, serum creatinine, and waist circumference. * 0.01 < *p* < 0.05; ** *p* < 0.01.

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
