# Peer review of "Central But Not General Obesity Is Positively Associated with the Risk of Hyperhomocysteinemia in Middle-Aged Women"

_nutrients, 2019, doi:10.3390/nu11071614_

Reviewer 1 Report

Clever hypothesis that is highly relevant in the scope of obesity and vascular related diseases - Large amount of subjects - Statistics are correct and support the hypothesis - Importance of the conclusions that bring to light additional hypotheses - Well written paper

This is an interesting study. The limitations are well described and are not empowering the interest of the study.

Author Response

We have read the manuscript thoroughly and modified potential spelling errors.

Reviewer 2 Report

This is a very well documented and well executed study investigating the relationships between hyperhomocysteinemia and obesity in middle-aged women. I only have minor recommendations for this manuscript.

There are minor grammatical errors throughout the manuscript, including the abstract. For example, the second sentence in the abstract.

In the first paragraph of the introduction, please spell out BMI and WC as that is the first time those acronyms are used in the main manuscript.

The statistical analysis section appears to be missing percentages for alcohol drinking and smoking. I am assuming the x's are placeholders.

Table 1 appears to have hidden text in many rows including the column headers. Please ensure that the final copy of the table is fully expanded.

Please refer to the journal provided author instructions for proper formatting of all references. Inconsistencies are observed in capitalization of article titles, abbreviations of journal names (should be fully written out).

Author Response

1. English language and style are fine/minor spell check required.

Response: We have read the manuscript thoroughly and modified potential spelling errors.

2. There are minor grammatical errors throughout the manuscript, including the abstract. For example, the second sentence in the abstract.

Response: We have modified the second sentence in the abstract as follows:

“However, evidence on the association between obesity and Hcy concentration was conflicting.”

3. In the first paragraph of the introduction, please spell out BMI and WC as that is the first time those acronyms are used in the main manuscript.

Response: We have spelled out BMI and WC in the first paragraph as suggested.

4. The statistical analysis section appears to be missing percentages for alcohol drinking and smoking. I am assuming the x's are placeholders.

Response:  Thank you for this. We have provided the percentages for alcohol drinkingand smoking as suggested. 

 “Smoking (0.28%) and regular alcohol drinking (0.73%) were not common among the women subjects and were not considered in the analysis”.

5. Table 1 appears to have hidden text in many rows including the column headers. Please ensure that the final copy of the table is fully expanded.

Response: Thank you and we have revised the manuscript to make sure all tables were presented properly.

6. Please refer to the journal provided author instructions for proper formatting of all references. Inconsistencies are observed in capitalization of article titles, abbreviations of journal names (should be fully written out).

Response: We have modified all references according to the instructions for authors.
